# Quantitative Microvascular Change Analysis Using a Semi-Automated Algorithm in Macula-on Rhegmatogenous Retinal Detachment Assessed by Swept-Source Optical Coherence Tomography Angiography

**DOI:** 10.3390/diagnostics14070750

**Published:** 2024-04-01

**Authors:** Pablo Díaz-Aljaro, Xavier Valldeperas, Laura Broc-Iturralde, Nevena Romanic-Bubalo, Ignacio Díaz-Aljaro, Zhongdi Chu, Ruikang K. Wang, Javier Zarranz-Ventura

**Affiliations:** 1Department of Ophthalmology, Hospital Universitari Germans Trias i Pujol, 08916 Badalona, Spain; pablodiazaljaro@gmail.com (P.D.-A.); laurabroc@gmail.com (L.B.-I.); nevenaromanic@gmail.com (N.R.-B.); 2Department of Surgery, Universitat Autònoma de Barcelona (UAB), 08035 Barcelona, Spain; 3Department of Ophthalmology, Hospital Clínico Universidad de Chile, Santiago 8380456, Chile; diazignacio3@gmail.com; 4Department of Bioengineering, University of Washington, Seattle, WA 98195-5061, USA; zhongdichu@gmail.com (Z.C.); wangrk@uw.edu (R.K.W.); 5Department of Ophthalmology, Hospital Clínic de Barcelona, 08036 Barcelona, Spain; jzarranz@hotmail.com

**Keywords:** rhegmatogenous retinal detachment, microvascular changes, optical coherence tomography angiography

## Abstract

Purpose: The purpose of this study was to objectively evaluate the longitudinal changes observed in the retinal capillaries in eyes with macula-on rhegmatogenous retinal detachment (RRD), assessed with optical coherence tomography angiography (OCTA), and to assess the role of these microvascular measures as potential biomarkers of postoperative visual outcomes. Methods: This was a prospective, longitudinal study conducted on consecutive patients who underwent 25 G pars plana vitrectomy for primary RRD. The vessel area density (VAD), vessel skeleton density (VSD), and vessel diameter index (VDI) were assessed in the superficial (SCP) and deep (DCP) capillary plexuses. Results: Twenty-three eyes with macula-on RRD were included in the study. The mean preoperative VDI, VAD, and VSD of the RRD eye were 18.6 ± 1.1, 0.43 ± 0.02, and 0.17 ± 0.01 in the SCP; and 18.6 ± 0.6, 0.38 ± 0.03, and 0.15 ± 0.01 in the DCP, respectively. At month 6, eight (34.8%) eyes achieved a best-corrected visual acuity (BCVA) gain ≥ 0.1 LogMAR. A preoperative VDI > 17.80, VSD > 0.170, and VDI-change > −0.395 in the SCP; and VDI > 18.11 and VDI change > −0.039 in the DCP were predictors of achieving a BCVA-gain ≥ 0.1 LogMAR. Conclusions: The preoperative VDI in SCP and DCP, the preoperative VSD in SCP, and their changes from preoperative values were predictors of visual outcomes. A quantitative OCTA analysis of capillary density and morphology may represent a potential predictive tool for surgical outcomes in RRD. The results reported in this study suggest that OCTA is a useful ancillary test for clinical disease management.

## 1. Introduction

Rhegmatogenous retinal detachment (RRD) is caused by a retinal tear or break allowing vitreous fluid to go through the subretinal space, leading to the separation of the neurosensory retina from the retinal pigment epithelium [1]. RRD often causes severe visual loss, especially when it involves the macula, and is currently considered one of the most common ophthalmological emergencies requiring immediate surgery [1,2].

Different therapeutic strategies, including pneumatic retinopexy, vitrectomy; scleral buckling; or a combination of both, are currently available for treating RRD [1,3,4]. Among them, pars plana vitrectomy (PPV) is an established and effective procedure, which is routinely considered as first choice in many cases [5].

Despite the increased rate of successful anatomical reattachment after surgery which is reported as near as 90%, a substantial proportion of patients may have limited functional results, with a wider range of outcomes for the final visual acuity [5,6,7,8,9]. Postoperative complications such as macular edema, epiretinal membrane, persistent subretinal fluid, and disruption of the inner segment/outer segment (IS/OS) are well-known causes of incomplete visual acuity improvement [10,11,12].

Different optical coherence tomography (OCT) biomarkers, including integrity of the external limiting membrane (ELM)/ellipsoid zone (EZ), cone interdigitation zone (CIZ), or foveal bulge, have been identified as predictors of RRD-surgery outcomes [13,14].

Optical coherence tomography angiography (OCTA) is a non-invasive tool that allows for ophthalmologists to assess the vascular system of the macula without any dye injection [15,16]. OCTA detects the motion of erythrocytes through a series of technical improvements, in speed and sensitivity, in the OCT imaging platform [17].

Although there seems to be an association between OCTA parameters and final best-corrected visual acuity (BCVA) in patients who underwent macula-off RRD surgery [18], the current evidence shows controversial results about the relationship between OCTA findings and RRD-surgery visual outcomes [19,20,21,22,23,24].

New developments in OCTA technology have improved the assessment of deep retinal microvasculature in vivo. Newer swept-source OCTA (SS-OCTA) technology uses longer wavelength and scanning speeds, which allow for increased penetration and faster image acquisition, enabling the capture of larger scan areas with improved image quality [17,25]. We introduce a semi-custom automated algorithm designed to assess OCTA images, extracting derived parameters such as vessel density and morphology in a layer-specific manner, enabling a three-dimensional interpretation of the scans in the selected region of interest, depending on the boundaries of the segmentation slab [26].

This study aimed to prospectively evaluate the OCTA metric changes observed in the retinal capillaries in macula on RRD eyes, followed up longitudinally for 6 months post-surgery. Additionally, we also assess the role of these microvascular measures as biomarkers of disease severity to investigate the potential use of OCTA as a predictive tool for postoperative visual outcomes.

## 2. Methods

### 2.1. Study Design

Prospective, longitudinal study conducted on consecutive patients who underwent 25 G PPV for primary RRD between May 2020 and June 2021 at the Ophthalmology Department of Hospital Universitari Germans Trias i Pujol, Barcelona, Spain. This study protocol was approved by the Ethics Committee of the Hospital Universitari Germans Trias i Pujol (OFT-AOCT-2018-01. Ref CEI PI-18/159) and adhered to the tenets of the current version of the Declaration of Helsinki and the Good Clinical Practice/International Council for Harmonization Guidelines. Written informed consent was obtained from all patients before participation in the study. To ensure patients confidentiality, any information that could lead to an individual being identified was encrypted or removed, as appropriate.

### 2.2. Study Participants and Selection Criteria

This study included consecutive patients, aged ≥ 18 years, with a clinical diagnosis of primary RRD and macula-on, who underwent a 25 G PPV between May 2020 and June 2021. Exclusion criteria included a high refractive error (spherical equivalent more than ±6), high myopia (axial length ≥ 26.0 mm), an axial length difference of more than 0.3 mm between the eyes, concomitant ocular diseases (diabetic retinopathy, glaucoma, age-related macular degeneration, uveitis, retinal vascular disease, or epiretinal membrane in either eye), prior vitreoretinal surgery, or prior retinal detachment in either eye. Patients with previous or postoperative important media opacities were also excluded because of inadequate OCTA acquisition and the low reliability of parameter measurements (Figure 1).

Patients underwent a complete ophthalmological examination of the affected (study eye) and the fellow eye, including BCVA measurement, using ETDRS (Early Treatment Diabetic Retinopathy Study) 2 m charts and a full battery of retinal imaging that included retinography, OCT, and OCTA (DRI OCT Triton; Topcon Corporation, Tokyo, Japan), before and 1, 3, and 6 months after surgery. Axial length (AL) measurement was obtained using noncontact partial coherence laser interferometry (IOL Master 500, version 3.01; Carl Zeiss Meditec, Jena, Germany).

#### 2.2.1. Surgical Technique

All patients underwent PPV performed under retrobulbar anesthesia by two different experienced high-volume surgeons (XV and LB). Standard three-port 25-gauge PPV without phacoemulsification was performed using the Alcon Constellation system (Alcon Laboratories, Inc., Fort Worth, TX, USA); no external limiting membrane peeling was performed, and SF6 and C3F8 gases were used as tamponades. Retinal reattachment was defined as the complete disappearance of subretinal fluid and flattening of the retinal breaks after gas reabsorption. No macular edema, epiretinal membrane, or significant cataract development was detected at any timepoint after surgery.

#### 2.2.2. Optical Coherence Tomography Angiography Image-Capture Protocol

OCTA centered in the macular area (DRI OCT Triton; Topcon Corporation, Tokyo, Japan) was performed in both eyes preoperatively and at month 1, 3, and 6 after surgery, respectively. For each eye, a 3 mm × 3 mm fovea-centered OCTA scan was performed; the SS-OCT and OCTA images were interpreted using the image viewer (ImageNet 6 Version 1.20.11109; Topcon Corporation), and an analysis was conducted on the superficial capillary plexus (SCP) and the deep capillary plexus (DCP). Automatic segmentation of the SCP and DCP was automatically performed by the built-in software (Figure 2).

Projection artifacts present in DCP were removed using a validated algorithm [27]. OCTA images in all cases were reviewed for image quality, and manual adjustment segmentation of the DCP was performed, when necessary, before proceeding with the vascular density analysis. Cases with distortion of the retinal architecture that prevented accurate segmentation, for example, motion artifacts, were excluded from the analysis. Central subfield thickness (CST) was assessed by the same OCT system, using the retina map mode.

A custom semiautomated algorithm was used to quantify several advanced parameters of retinal vascular density and morphology that are described elsewhere [28]. These parameters included vessel area density (VAD), vessel skeleton density (VSD), and vessel diameter index (VDI). To quantify these parameters, the grayscale 2D En-Face SS-OCTA image was first converted into an 8-bit image (586 × 585 pixels), encompassing a 3 × 3-mm^2^ area around the fovea (1 pixel = 5.13 × 5.13 um^2^). This image was converted into a binary image and a skeletonized image by using a three-way combined method consisting of a global threshold, hessian filter, and adaptive threshold in MATLAB (R2013b; MathWorks, Inc., Natick, MA, USA). These parameters are designed to quantitatively characterize capillary density (VAD and VSD) and capillary morphology (VDI) since both density and morphology are known variables in the development and progression of retinal vascular disease. VAD is derived from the binarized OCTA image and quantifies the percentage of the angiogram area with detectable perfusion. The binarized image can be reduced to a skeletonized representation of retinal vessels by reducing the width of each vessel segment to one pixel. VSD is then derived from this skeletonized image and represents the absolute linear distance (length) of blood vessels in the image. Therefore, VSD is representative of the length of the entire retinal vascular network independent of vessel caliber [28,29,30,31]. VDI is derived from both binarized and skeletonized images and quantifies the average vascular caliber (vessel diameter). These parameters (VDI, VSD, and VAD) were previously demonstrated to have very good repeatability and reliability [28,29,30,31]. In addition, the contralateral eye was assessed for evaluating potential correlations.

### 2.3. Main Outcome Measurements

The primary endpoints were the mean VDI, VAD, and VSD changes from preoperative to month 6. Secondary endpoints included the differences in VDI, VAD, and VSD between the study and the contralateral eye at the different timepoint measures; the correlation of the OCTA vascular preoperative parameters between the study and contralateral eye; and the predictive factors associated with functional success (BCVA improvement ≥ 0.1 LogMAR at month 6).

### 2.4. Statistical Analysis

The statistical software package (MedCalc^®^ Statistical Software version 20.216. MedCalc Software Ltd., Ostend, Belgium; https://www.medcalc.org; accessed on 13 May 2023) was used for the analysis. Descriptive statistics mean ± standard deviation (SD), median (interquartile range, IqR), and number (percentage) were used as appropriate. Data were tested for normal distribution using a Shapiro–Wilk test. The changes in VDI, VAD, and VSD over the course of the study were evaluated with the repeated-measures ANOVA or a Friedman’s two-way analysis test, as appropriate. The Mann–Whitney U test was used to compare baseline and evolutive continuous variables, from preoperative to month 6, between study group (eyes with RRD) and fellow eyes. A logistic regression model was used to estimate and test factors for their association with functional success. A backward strategy was adopted, with a statistically significant cutoff for variable screening of 0.05. Factors associated with BCVA gain ≥ 0.1 LogMAR in the univariate analysis at *p* < 0.1 were included in the multivariate analysis.

Receiver operating characteristic (ROC) curves were constructed to assess the sensitivity and specificity of OCTA variables for the prediction of BCVA improvement ≥ 0.1. LogMAR. The area under the curve was calculated for each variable with the corresponding upper and lower asymmetric 95% confidence interval and the *p*-value for difference from 0.5 as the result for a variable with no predictive capacity.

The optimal point on the ROC curves that provided best discrimination (the point on the curve nearest the top left of the graph) was determined for the VDI, VAD, and VSD in the SCP and the DCP. Categorical variables were compared using a Chi-square test and Fisher’s exact test, as appropriate. A *p*-value of less than 0.05 was considered significant.

## 3. Results

### 3.1. Preoperative Demographic and Clinical Characteristics

Twenty-three eyes from 23 patients were included in the study. The mean age was 61.9 ± 7.5 years, and eight (34.8%) were women. In all, 18 (78.3%) eyes did not have preoperative PVR, the extension of the RD was ≤2 quadrants in 22 (95.7%) eyes, 5 (21.7%) were pseudophakic, and phakic patients 18 (78.3%) did not require cataract surgery during follow-up. Table 1 summarizes the main demographic and clinical characteristics of the study sample.

### 3.2. OCTA Parameters

In the superficial capillary plexus, the mean VDI, VAD, and VSD of the study eye were 18.6 ± 1.1, 0.43 ± 0.02, and 0.17 ± 0.01, respectively. Meanwhile, in the DCP, the mean VDI, VAD, and VSD of the study eye were 18.6 ± 0.6, 0.38 ± 0.03, and 0.15 ± 0.01.

As compared to the fellow eye, preoperative VDI in the SCP was significantly lower in the study eye (Hodges–Lehmann median difference, 0.51; 95%CI, 0.02 to 1.15; *p* = 0.04). Meanwhile, in the DCP, the preoperative VDI (Hodges-Lehmann median difference, 0.34; 95% CI, 0.01 to 0.67; *p* = 0.04) and VAD (Hodges–Lehmann median difference, 0.02; 95% CI, 0.00 to 0.04; *p* = 0.04) were significantly lower in the study eye (Table 2). In the eye diagnosed with RRD, the mean OCTA parameters did not show significant changes throughout the study follow-up for SCP or DCP (Table 3).

The preoperative and month-1 VDI in both the SCP (*p* = 0.04 and *p* = 0.02, respectively) and DCP (*p* = 0.04 and *p* = 0.01, respectively) were significantly lower in the RRD eye. Additionally, in the DCP, the preoperative and month-6 VAD were significantly lower in the RRD eye (*p* = 0.04 and *p* = 0.03, respectively). No significant differences were observed in other OCTA parameters between the RRD and the fellow eyes (Figure 3).

We assessed the correlation of OCTA parameters between the eyes with RRD and the fellow eyes (Appendix A). No significant correlations were observed in any of the vascular parameters measured with OCTA, either in the SCP or in the DCP, between the eye with RRD and the fellow eye.

### 3.3. Predictive Factors for Functional Outcome

The mean BCVA did not show significant changes from preoperative (0.20 ± 0.29) to month 6 (0.30 ± 0.39) (mean difference, 0.10; 95%CI, −0.04 to 0.25; *p* = 0.14). Nevertheless, at month 6, eight (34.8%) eyes achieved a BCVA gain ≥ 0.1 LogMAR.

In the univariate analysis, preoperative factors significantly associated with the probability of achieving a BCVA improvement ≥ 0.1 LogMAR were VDI (odds ratio (OR), 2.64; 95%CI, 1.01 to 6.89; *p* = 0.04) and VSD (OR, 2.97; 95%CI, 1.19 to 6.51; *p* = 0.03) in the SCP, and VDI in the DCP (OR, 5.88; 95%CI, 1.01 to 35.14; *p* = 0.04). Additionally, the mean change from preoperative to month-6 values of the VDI (odds ratio (OR), 6.67; 95%CI, 1.09 to 40.79; *p* = 0.04) and of the VSD (OR, 3.98; 95%CI, 1.02 to 11.37; *p* = 0.04) in the SCP were significantly associated with a BCVA gain ≥ 0.1 LogMAR. None of the variables showed statistically significant values in the multivariate analysis (Table 4)

### 3.4. ROC Curves

To further investigate the predictive capacity of the OCTA parameters measured in the SCP and the DCP on the visual outcomes, ROC curves were calculated for all preoperative OCTA variables and variable changes at the month-6 timepoint (Table 5). In the SCP, the area under the curve was significantly different from 0.5 for the preoperative VDI and VSD, and for the VDI change from preoperative to month-6 values. Meanwhile, in the DCP, the area under the curve was significantly different from 0.5 for the preoperative VDI and the VDI change from preoperative to month-6 values. The optimal points (i.e., the points that provide best the discrimination for both sensitivity and specificity) were VDI > 17.80, VSD > 0.170, and VDI change > −0.395 in the SCP; and in the DCP, the optimal points were VDI > 18.11 and VDI change > −0.039.

## 4. Discussion

This study reports significant associations between OCTA parameters and the 6-month postoperative visual outcome after macula-on RRD surgery and highlights the predictive value of different preoperative baseline OCTA variables on the final visual outcomes in these surgical patients. The findings described in this study suggest that OCTA may have a potential role as a predictor of the final visual outcome and therefore advocate for its use in the preoperative assessment of macula on RRD cases.

Microvascular alterations in retinal and choroidal circulation occurring in RRD eyes, even without macular involvement, have long been studied [19,20,21,22,23,24]. These macular microvascular changes, together with the photoreceptor alterations, are supposed to be responsible for the lack of vision improvement, despite good anatomical results [32,33].

In macula-on RRD eyes, we did not observe significant changes in any of the OCTA parameters analyzed, either SCP or DCP, throughout the follow-up. According to our results, greater preoperative VDI or VSD in the SCP, and VDI in the DCP were identified as biomarkers of achieving good visual outcome. Additionally, a positive association was observed for VDI in both SCP and DCP, and the greater VDI, the greater the probability of achieving good visual outcomes.

We found significantly lower preoperative values of the VDI (both SCP and DCP) and VAD (DCP) in the RRD eyes compared to the fellow non-operated eyes of the patients. These lower VDI values may be explained by several reasons, including a potential local vasoconstriction component which may be present in these patients in response to the vasoactive changes induced by the release of pigmented cells through the retinal tear, or a transient status of homeostatic changes in the vitreous cavity, among other potential explanations which could be related to the acute event. These results differ from those reported by Yoshikawa et al. [20], who found no changes in the SCP after surgery in RRD eyes without macular involvement but may be in line with the results reported by Woo et al. [18] that, like our study, found more vascular alterations in the DCP, suggesting a greater vulnerability of DCP to tissue hypoxia.

Interestingly, only month-1 VDI, in both SCP and DCP, and month-6 VAD in DCP showed significant differences between RRD eyes and fellow eyes after surgery, while the other OCTA parameters did not. These results partially agree with those of the meta-analysis published by Chen et al. [34], who reported no microvascular differences between the eyes with RRD and the fellow eyes after surgery. Similarly, Bonfiglio et al. [33] found lower parafoveal deep vessel density, but they did not find significant differences in the foveal avascular zone (FAZ) compared to fellow eyes after surgery. They also reported that, in macula-on RRD eyes, the final BCVA was related to the FAZ area and deep vessel density [33]. Additionally, a negative correlation between the final postoperative BCVA and FAZ area in both SCP [18,21] and DCP [18] was observed. A recent study published by Machairoudia et al. [35] found significant changes in the foveal avascular zone in the superficial and deep capillary plexus at 6 months between operated eyes compared to fellow eyes with a moderate correlation with BCVA, and no significant differences in vessel density were observed.

As far as we know, this is the first study investigating the association between these advanced OCTA-derived vessel parameters and postoperative visual outcomes in successfully treated macula-on RRD eyes. We observed in the univariate analysis that, in the SCP, the preoperative VDI and VSD and the 6-month changes in both parameters were predictors of good visual outcome, and in the DCP, this remained significant for VDI. We hypothesize that these changes may be explained at least in part due to the different vascular and inflammatory mediators that are released by the Müller cells that are activated in eyes with RRD [32,36]. These events may induce microarchitectural modifications that may impair retinal ganglion cells and photoreceptors’ functionality, which could eventually lead to the subsequent lack of visual improvement [37,38].

A clinically relevant finding of this study is the predictive value of the preoperative status of the retinal microvasculature. A previous study by Chatziralli et al. [39] demonstrated that, in macula-off detachments, a duration longer than one week was found to be a detrimental prognostic factor for visual outcome. However, our analysis, both univariate and multivariate, did not find support to confirm this association. We found that the eyes that, according to the OCTA, presented less preoperative vasoconstriction, had a greater probability of achieving better visual outcomes. This raises the interesting hypothesis that those eyes with a relatively preserved vessel diameter may reflect a lower degree of subclinical inflammation, which may be associated with a lower preoperative vascular impairment status and, therefore, a greater probability of recovering visual acuity postoperatively.

The fact that not only the preoperative microvascular parameters but also the changes that occurred in these parameters are related to the visual results highlights the importance of evaluating the state of the microcirculation in eyes with RRD.

This study presents a series of limitations. First, it includes a single-center cohort of eyes operated by two experienced surgeons, with a relatively small sample size and a short follow-up period. Having said that, these limitations are partially compensated by the prospective study design and the lack of variability in the surgical technique often seen in larger multicenter studies. Another limitation is the technical difficulty of obtaining adequate OCTA scans in eyes with RRD, in which fixation is often impaired even in cases of macular sparing, or the higher frequency of projection artifacts related to vitreous cavity opacities compared to healthy eyes that prevent the adequate segmentation of the slabs, which may have particularly affected the metrics obtained in the DCP. To overcome the latter limitation, manual corrections of the DCP segmentation slab were performed when necessary, adding a potential confounding factor to generalize the results described in the research setting of this study to the automatic segmentations offered in the commercial version of the OCTA.

In conclusion, this study reports that OCTA provide relevant information about the preoperative status of the retinal microvasculature and its longitudinal changes throughout the follow-up in RRD eyes, with significant differences in preoperative vessel morphology compared to fellow eyes. Our findings suggest that preoperative OCTA-derived parameters such as VDI and VSD and their changes from preoperative values in the SCP, or VDI in the DCP, could be considered predictors of visual outcomes in macula on RRD cases. If validated in future studies, OCTA could play a significant role in the preoperative assessment of these surgical cases.

## Figures and Tables

**Figure 1 diagnostics-14-00750-f001:**
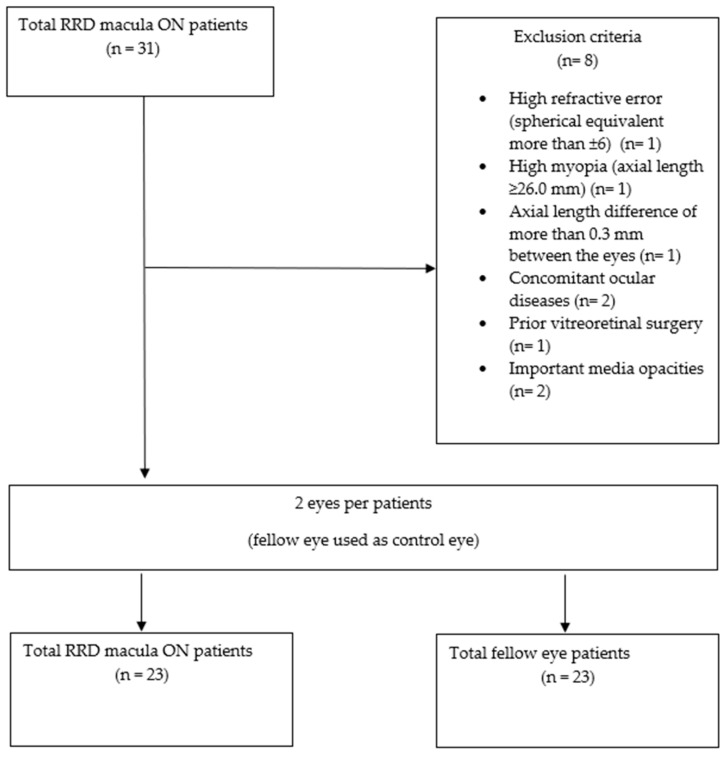
Flowchart showing the selection of patients in this study.

**Figure 2 diagnostics-14-00750-f002:**
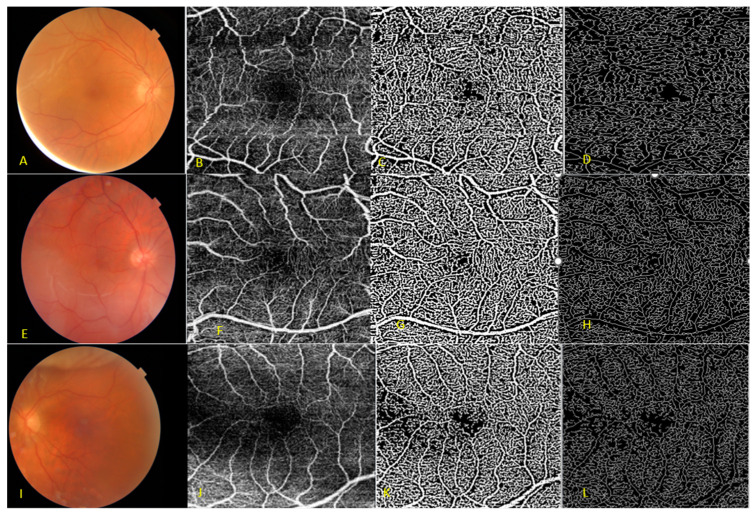
Visual illustrations that serve to depict the quantitative analysis algorithm in three cases at the superficial capillary plexus. (**A**,**E**,**I**) Color fundus photography of macula-on rhegmatogenous retinal detachment. (**B**,**F**,**J**) Original optical coherence tomography angiography (OCTA) En-face image. (**C**,**G**,**K**) Vessel area map, a binary vasculature image created through the Hessian filter and adaptive thresholding. This image is employed for the quantification of VAD and VDI. (**D**,**H**,**L**) Representation of the vessel skeleton map, generated by iteratively removing pixels from the outer perimeter of the vessel area map until only one pixel remains in the width direction of the vessels. This image is utilized in the quantification of VSD and VDI.

**Figure 3 diagnostics-14-00750-f003:**
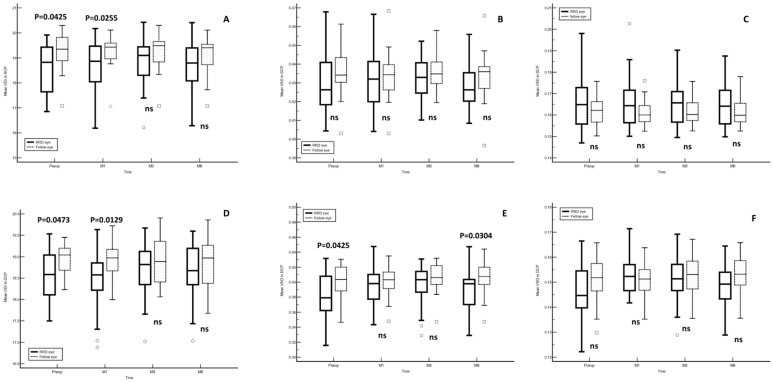
Mean optical coherence tomography angiography (OCTA) vascular parameters. The *p*-values were calculated with the Mann–Whitney U test. (**A**) VDI in SCP, (**B**) VAD in SCP, (**C**) VSD in SCP, (**D**) VDI in DCP, (**E**) VAD in DCP and (**F**) VSD in DCP. RRD, rhegmatogenous retinal detachment; VDI, vessel diameter index; VAD, vessel area density; VSD, vessel skeletal density; SCP, superficial capillary plexus; DCP, deep capillary plexus; ns, not significant.

**Table 1 diagnostics-14-00750-t001:** Overview of the main demographic and clinical characteristics of the study sample.

	Study Population (*n* = 23 Eyes)
Age, years	
Mean ± SD	61.9 ± 7.5
Median (IqR)	60.9 (56.7 to 65.8)
Sex, *n* (%)	
Women	8 (34.8)
Men	15 (65.2)
Systemic diseases, *n* (%)	
DM	3 (13.0)
HBP	6 (26.1)
Dyslipidemia	7 (30.4)
Smoker	1 (4.8)
Other	4 (17.4)
Eye, *n* (%)	
Right	16 (69.6)
Left	7 (30.4)
PVR, *n* (%)	
No	18 (78.3)
Degree A	3 (13.0)
Degree B	2 (8.7)
Time of RRD, *n* (%)	
<7 days	12 (52.2)
≥7 days	11 (47.8)
Extension, *n* (%)	
1 quadrant	12 (52.2)
2 quadrants	10 (43.5)
3 quadrants	0 (0.0)
4 quadrants	1 (4.3)
Lens status, *n* (%)	
Phakic	18 (78.3)
Pseudophakic	5 (21.7)
IOP *, mmHg	
Mean ± SD	15.8 ± 2.6
Median (IqR)	16.0 (14.0 to 18.0)
Axial length **, mm	
Mean ± SD	25.7 ± 1.6
Median (IqR)	25.7 (24.3 to 26.4)
BCVA, logMAR	
Mean ± SD	0.20 ± 0.29
Median (IqR)	0.1 (0.0 to 0.28)
CRT, µm	
Mean ± SD	296.4 ± 25.3
Median (IqR)	293.0 (266.2 to 315.0)

* Information available only in 19 eyes. ** Information available only in 14 eyes. SD, standard deviation; IqR, interquartile range; DM, diabetes mellitus; HBP, systemic high blood pressure; PVR, proliferative vitreoretinopathy; IOP, intraocular pressure.

**Table 2 diagnostics-14-00750-t002:** A comparison of the superficial capillary and deep capillary plexuses optical coherence tomography angiography (OCTA) parameters between the eyes with rhegmatogenous retinal detachment (RRD) and the fellow eyes.

	**Superficial Capillary Plexus (*n* = 23)**		
	**RRD**	**Fellow Eyes**	***p* ^a^**
VDI			0.0425
Mean ± SD	18.6 ± 1.1	19.2 ± 0.7	
Median (IqR)	18.8 (17.6–19.4)	19.3 (18.9–19.8)	
VAD			0.2380
Mean ± SD	0.43 ± 0.02	0.44 ± 0.01	
Median (IqR)	0.43 (0.42–0.44)	0.43 (0.43–0.44)	
VSD			0.3596
Mean ± SD	0.17 ± 0.01	0.16 ± 0.01	
Median (IqR)	0.17 (0.16–0.17)	0.16 (0.16–0.17)	
	**Deep Capillary Plexus (*n* = 23)**		
	**RRD**	**Fellow Eyes**	***p* ^a^**
VDI			0.0473
Mean ± SD	18.6 ± 0.6	19.0 ± 0.4	
Median (IqR)	18.6 (18.1–190.0)	19.0 (18.7–19.29)	
VAD			0.0425
Mean ± SD	0.38 ± 0.03	0.40 ± 0.02	
Median (IqR)	0.38 (0.36–0.41)	0.40 (0.39–0.42)	
VSD			0.1305
Mean ± SD	0.15 ± 0.01	0.15 ± 0.01	
Median (IqR)	0.15 (0.14–0.16)	0.15 (0.15–0.16)	

^a ^Mann–Whitney U-test. Study eyes—eyes with RRD; SD—standard deviation; IqR—interquartile range; VDI—vessel diameter index; VAD—vessel area density; VSD—vessel skeletal density.

**Table 3 diagnostics-14-00750-t003:** Overview of the preoperative and follow-up optical coherence tomography angiography (OCTA) parameters in the superficial capillary and deep capillary plexuses in the eyes with rhegmatogenous retinal detachment (RRD). Statistical significance was calculated with the Friedman test.

	**Superficial Capillary Plexus (*n* = 23)**				
	**Preoperative**	**Month 1**	**Month 3**	**Month 6**	** *p* **
VDI					0.990
Mean ± SE	18.6 ± 0.2	18.6 ± 0.2	18.8 ± 0.2	18.7 ± 0.2	
95%CI	18.1–19.1	18.1–19.1	18.4–19.2	18.2–19.1	
VAD					0.6704
Mean ± SE	0.43 ± 0.00	0.43 ± 0.00	0.43 ± 0.00	0.43 ± 0.00	
95%CI	0.42–0.44	0.43–0.44	0.43–0.44	0.42–0.43	
VSD					0.4781
Mean ± SE	0.17 ± 0.00	0.17 ± 0.00	0.17 ± 0.00	0.16 ± 0.00	
95%CI	0.16–0.17	0.16–0.17	0.16–0.17	0.16–0.17	
	**Deep Capillary Plexus (*n* = 23)**				
	**Preoperative**	**Month 1**	**Month 3**	**Month 6**	** *p* **
VDI					0.9586
Mean ± SE	18.6 ± 0.13	18.5 ± 0.2	18.7 ± 0.1	18.6 ± 0.1	
95%CI	18.3–18.9	18.2–18.8	18.5–19.0	18.4.18.9	
VAD					0.0735
Mean ± SE	0.38 ± 0.01	0.40 ± 0.01	0.40 ± 0.01	0.39 ± 0.01	
95%CI	0.37–0.40	0.39–0.41	0.38–0.41	0.38–0.40	
VSD					0.1821
Mean ± SE	0.15 ± 0.00	0.15 ± 0.00	0.15 ± 0.00	0.15 ± 0.00	
95%CI	0.14–01.5	0.15–0.16	0.15–0.16	0.15–0.15	

SE, standard error; 95%CI, 95% confidence interval; VDI, vessel diameter index; VAD, vessel area density; VSD, vessel skeletal density.

**Table 4 diagnostics-14-00750-t004:** Univariate and multivariate analysis to evaluate the potential factors for achieving a best-corrected visual acuity improvement ≥ 0.1 at month 6.

Variable	Univariate		Multivariate	
	OR (95%CI)	*p*	OR (95%CI)	*p*
Age ^a^	0.96 (0.85 to 1.09)	0.4581		
Sex				
Ref Men				
Women	0.16 (0.02 to 1.68)	0.1270		
DM				
Ref No				
Yes	0.93 (0.07 to 12.14)	0.9549		
HBP				
Ref No				
Yes	0.92 (0.13 to 6.56)	0.9309		
Dyslipidemia				
Ref No				
Yes	0.67 (0.10 to 4.58)	0.6800		
RD extension				
Ref 1 quadrant				
≥2 quadrants	1.14 (0.21 to 6.379	0.8789		
Time of RRD				
Ref < 7 days				
≥7 days	6.0 (0.87 to 41.22)	0.0684	5.07 (0.62 to 41.44)	0.1299
PVR				
Ref No				
Yes	3.90 (0.50 to 30.76)	0.1965		
Axial length ^b^	0.80 (0.28 to 2.27)	0.6766		
Preop VDI SCP ^c^	2.64 (1.01 to 6.89)	0.0468	2.48 (0.89 to 6.90)	0.0810
Preop VAD SCP ^d^	1.56 (0.13 to 5.92)	0.1067		
Preop VSD SCP ^d^	2.97 (1.19 to 6.51)	0.0350	2.76 (0.90 to 7.03)	0.0838
Preop VDI DCP ^c^	5.88 (1.01 to 35.14)	0.0493	4.89 (0.76 to 31.60)	0.0546
Preop VAD DCP ^d^	5.63 (0.03 to 78.11)	0.8481		
Preop VSD DCP ^d^	4.43 (0.14 to 11.29)	0.2435		
Mean change ^1^ VDI SCP ^e^	6.67 (1.09 to 40.79)	0.0401	6.23 (0.93 to 52.16)	0.0584
Mean change ^1^ VAD SCP ^d^	2.50 (0.01 to 16.27)	0.9742		
Mean change ^1^ VSD SCP ^d^	3.98 (1.02 to 11.37)	0.0410	3.75 (0.91 to 15.08)	0.0880
Mean change ^1^ VDI DCP ^e^	6.37 (0.78 to 51.82)	0.0833	5.99 (0.43 to 40.61)	0.2993
Mean change ^1^ VAD DCP ^d^	5.17 (0.14 to 29.72)	0.2639		
Mean change ^1^ VSD DCP ^d^	4.39 (0.08 to 19.27)	0.6552		

^a^ Per year older. ^b^ Per mm longer. ^c^ Per unit greater. ^d^ Per 0.01 units greater. ^e^ Per 0.1 units reduced. ^1^ Mean change from preoperative to month-6 values. DM, diabetes mellitus; HBP, high blood pressure; RD, retinal detachment; PVR, proliferative vitreoretinopathy; Preop, preoperative; SCP, superficial capillary plexus; DCP, deep capillary plexus; VDI, vessel diameter index; VAD, vessel area density; VSD, vessel skeletal density.

**Table 5 diagnostics-14-00750-t005:** Receiver operating characteristic (ROC) curves of the different optical coherence tomography angiography (OCTA) variables with prediction of best corrected visual acuity improvement ≥ 0.1.

	**Superficial Capillary Plexus**						
	**Area (SE)**	**95%CI ^a^**	** *p* **	**Youden Index (95%CI) ^a^**	**Criterion**	**Sensitivity**	**Specificity**
VDI	0.742 (0.121)	0.446–0.917	0.0463	0.492 (0.243–0.683)	>17.80	86.67	62.50
VAD	0.692 (0.118)	0.433–0.883	0.1055	0.408 (0.217–0.600)	>0.423	87.50	53.33
VSD	0.792 (0.105)	0.500–0.942	0.0056	0.550 (0.217–0.800)	>0.170	75.00	80.00
VDI change *	0.833 (0.089)	0.600–0.958	0.0002	0.542 (0.230–0.742)	>−0.395	66.67	87.50
VAD change *	0.542 (0.152)	0.263–0.808	0.7834	0.367 (0.208–0.550)	≤−0.014	50.00	86.57
VSD change *	0.725 (0.137)	0.400–0.933	0.1002	0.500 (0.242–0.750)	≤−0.016	50.00	100.00
	**Deep Capillary Plexus**						
	**Area (SE)**	**95%CI ^a^**	** *p* **	**Youden Index (95%CI) ^a^**	**Criterion**	**Sensitivity**	**Specificity**
VDI	0.742 (0.112)	0.471–0.908	0.0306	0.492 (0.225–0.750)	>18.11	86.67	62.50
VAD	0.500 (0.132)	0.233–0.738	1.0000	0.200 (0.150–0.200)	>0.410	20.00	100.00
VSD	0.658 (0.118)	0.400–0.858	0.1791	0.333 (0.150–0.425)	>0.139	100.00	33.33
VDI change *	0.917 (0.057)	0.717–0.983	<0.0001	0.733 (0.400–0.867)	>−0.039	73.33	100.00
VAD change *	0.633 (0.120)	0.383–0.842	0.2673	0.400 (0.208–0.600)	>−0.027	40.00	100.00
VSD change *	0.508 (0.123)	0.275–0.750	0.9461	0.275 (0.200–0.333)	>−0.001	40.00	87.50

* Mean change from preoperative to month-6 values. ^a^ BC_a_ bootstrap confidence interval (10,000 iterations; random number seed, 9780). SE, standard error; 95%CI, 95% confidence interval; VDI, vessel diameter index; VAD, vessel area density; VSD, vessel skeletal density.

## Data Availability

Data available upon request from the corresponding author.

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
