# Peer review of "Quantitative Microvascular Change Analysis Using a Semi-Automated Algorithm in Macula-on Rhegmatogenous Retinal Detachment Assessed by Swept-Source Optical Coherence Tomography Angiography"

_diagnostics, 2024, doi:10.3390/diagnostics14070750_

Round 1

Reviewer 1 Report

Comments and Suggestions for Authors

The study has shown the relationship between OCTA features and prognosis of macula-on RRD. The authors finally showed the diagnostic power to predict postoperative VA.

The paper is well written and clearly shows the results of the study.

Minor comment

Several representative cases should be shown with FP, OCTA, and detailed measurements.

Author Response

Dear Reviewer,

Thank you for your valuable feedback. We have taken into account your suggestion regarding the inclusion of several representative cases showcasing FP, OCTA, and detailed measurements. We have incorporated these cases into the manuscript as per your recommendation.

We have also improved quality of figures and tables already exposed on previous manuscript

Best regards

Pablo Díaz-Aljaro

Reviewer 2 Report

Comments and Suggestions for Authors

I would like to congratulate the authors on their work. A few points that could further improve their manuscript are:

a.The authors describe the inclusion and exclusion criteria for participants. They could add a flowchart to present how many patients were initially approached, how many patients declined to take part, how many were not deemed eligible etc.

b.Please add more detail about the surgical technique used for the vitrectomy. Instead of 25 gauge PPV please add details, which tamponade agents were used, was ILM peeling performed etc. You can also add that the surgeons who performed the surgeries were experienced high volume surgeons.

c.One of the findings of the present study that need more attention is the fact that in univariate analysis the duration of detachment was associated with BCVA outcome, but this association was not achieved in the multivariate analysis. A previous study1 also found that in macula-off detachments the duration of 1 week was found as a poor prognostic factor for visual outcome.

d.The results are well written and supported by previous scientific evidence. The authors also mention that some microvascular indices in month 1 and month 6 differed and present some evidence to support this conclusion. They could also add another recent study that found significant changes in the foveal avascular zone in the superficial and deep capillary plexus2 which correlates moderately with BCVA, between operated and fellow eyes.

1Chatziralli I, Chatzirallis A, Kazantzis D, Dimitriou E, Machairoudia G, Theodossiadis G, Parikakis E, Theodossiadis P. Predictive Factors for Long-Term Postoperative Visual Outcome in Patients with Macula-Off Rhegmatogenous Retinal Detachment Treated with Vitrectomy. Ophthalmologica. 2021;244(3):213-217. doi: 10.1159/000514538. Epub 2021 Jan 19. PMID: 33465770.

2.Machairoudia G, Kazantzis D, Chatziralli I, Theodossiadis G, Georgalas I, Theodossiadis P. Microvascular changes after pars plana vitrectomy for rhegmatogenous retinal detachment repair: A comparative study based on gas tamponade agent. Eur J Ophthalmol. 2023 Dec 3:11206721231218656. doi: 10.1177/11206721231218656. Epub ahead of print. PMID: 38043935.

Author Response

Dear Reviewer,

Thank you for your insightful comments. We have addressed your suggestion by incorporating a flowchart in the manuscript. This flowchart outlines the patient recruitment process.

Regarding the surgical technique used for vitrectomy. In response, we have provided additional details in the manuscript, including the specific tamponade agents utilized and whether internal limiting membrane (ILM) peeling was performed. Furthermore, we have noted that the surgeries were conducted by experienced high-volume surgeons, as per your recommendation.

In addressing the discussion concerning the duration of detachment and its correlation with visual outcomes, we have cited the preceding study nº 1, which aligns with our findings and offers  contextualization to our results. Additionally, we have incorporated study numbered 2 into the discussion section, revealing notable alterations in microvascular indices and their relationship with visual outcomes. This supports our conclusions and further understanding of the  microvascular changes observed between operated and fellow eyes.

Best regards,

Pablo Díaz-Aljaro